# Comparative Genome Analyses Provide Insight into the Antimicrobial Activity of Endophytic *Burkholderia*

**DOI:** 10.3390/microorganisms12010100

**Published:** 2024-01-04

**Authors:** Jiayuan Jia, Shi-En Lu

**Affiliations:** Department of Biochemistry, Molecular Biology, Entomology and Plant Pathology, Mississippi State University, Mississippi State, MS 39762, USA; jj1748@msstate.edu

**Keywords:** comparative genomics, endophytic bacteria, *Burkholderia*, antimicrobial

## Abstract

Endophytic bacteria are endosymbionts that colonize a portion of plants without harming the plant for at least a part of its life cycle. Bacterial endophytes play an essential role in promoting plant growth using multiple mechanisms. The genus *Burkholderia* is an important member among endophytes and encompasses bacterial species with high genetic versatility and adaptability. In this study, the endophytic characteristics of *Burkholderia* species are investigated via comparative genomic analyses of several endophytic *Burkholderia* strains with pathogenic *Burkholderia* strains. A group of bacterial genes was identified and predicted as the putative endophytic behavior genes of *Burkholderia*. Multiple antimicrobial biosynthesis genes were observed in these endophytic bacteria; however, certain important pathogenic and virulence genes were absent. The majority of resistome genes were distributed relatively evenly among the endophytic and pathogenic bacteria. All known types of secretion systems were found in the studied bacteria. This includes T3SS and T4SS, which were previously thought to be disproportionately represented in endophytes. Additionally, questionable CRISPR-Cas systems with an orphan CRISPR array were prevalent, suggesting that intact CRISPR-Cas systems may not exist in symbiotes of *Burkholderia*. This research not only sheds light on the antimicrobial activities that contribute to biocontrol but also expands our understanding of genomic variations in *Burkholderia*’s endophytic and pathogenic bacteria.

## 1. Introduction

Plant-growth-promoting bacteria (PGPB) are a group of plant-associated bacteria that enhance growth and protect from disease. Various means are employed by PGPB to accomplish a plant-promoting effect, including the production of plant hormones; nitrogen fixation; phosphate solubilization; the synthesis and production of antimicrobial compounds, antibiotics, and enzymes; and competition with phytopathogenic microorganisms [1,2,3,4,5]. PGPB consist of several types of microorganisms, specifically endophytes residing inside of plant tissues, epiphytes that live externally on plants, and root-inhabiting rhizospheric bacteria. These bacteria are mutualistic symbionts and benefit the host by enhancing disease resistance, increasing tolerance to environmental stresses, and acting as biocontrol agents. Several researchers have documented how the colonization of healthy plant tissues by endophytes does not generate any symptoms of disease [6,7]. In addition, the proximity of endophytic bacteria may allow them to provide a greater positive influence on host plants when compared with epiphytic and rhizospheric bacteria, especially during critical and harsh conditions [8]. Studies have also demonstrated that endophytes promote growth in numerous plant hosts, demonstrating their potential as biofertilizers and biopesticides [9,10]. Based on the available research, it is likely that bacterial endophytes are involved in the production of bioactive compounds in conjunction with the promotion of plant growth and the suppression of disease [11,12]. There are more than 200 genera of bacteria that have been reported as endophytes, which include *Bacillus*, *Pseudomonas*, *Azoarcus,* and *Burkholderia* [7].

The bacterial genus *Burkholderia* encompasses an array of species with high genetic versatility and adaptability, allowing them to exist in diverse ecological niches such as soils and plants, as well as animal and human bodies [13]. In the *Burkholderia* genome database, there are currently 287 *Burkholder* strains with complete genomes [14]. The genome size of *Burkholderia* bacteria varies from 5.4 Mb to 9.7 Mb with up to three chromosomes and five plasmids [14]. The multiple replicons within the *Burkholderia* genome are important elements that influence the genetic factors that actualize their ecological diversity [15]. In addition, the high proportion of coding genes and rich insertion sequences that are present in the *Burkholderia* genome are often associated with metabolite production, on which their adaptability is founded [16].

Recently, significant advances have been made in the taxonomy of *Burkholderia* sensu lato. The *Burkholderia* sensu lato is divided into seven genera: *Burkholderia* sensu stricto, *Paraburkholderia*, *Caballeronia* [17], *Mycetohabitans*, *Trinickia* [18], *Robbsia* [19], and *Pararobbsia* [20]. However, significant uncertainty and flux remain in this large and heterogeneous taxon [21]. Within the *Burkholderia* sensu stricto, there are plant pathogens and opportunistic human pathogens responsible for pulmonary disease [22]. The species *Burkholderia plantarii* and *Burkholderia gladioli* are rice pathogens and are primarily responsible for sheath rot and seedling blight, respectively. Some strains of *Burkholderia cepacia*, *Burkholderia cenocepacia*, and *Burkholderia multivorans* can act as opportunistic human pathogens, causing respiratory complications for cystic fibrosis (CF) patients [23]. On the other hand, some *Burkholderia* sensu lato bacteria are beneficial species with plant-growth-promoting activities [24]. It is well established that *Burkholderia* produces various secondary metabolites, including compounds with antibacterial, antifungal, insecticidal, and herbicidal properties [25]. For example, *B. cepacia* may act as an effective biocontrol agent for Pythium-induced damping-off and *Rhizoctonia solani*-induced root rot in poinsettias [26]. While it is very probable that there are certain differences between pathogenic and endophytic groups at the genomic level, the genetic distinction between pathogenic and endophytic strains within the genus *Burkholderia* is unclear.

In this study, we analyzed the genetic relatedness of 18 endophytic and pathogenic strains within the genus *Burkholderia*. Comparative genome analysis revealed potential endophytic-behavior-related genes and a close relationship between the endophytic strains. Distinctive features between endophytic and pathogenic bacteria were identified by analyzing the production of antimicrobial compounds, virulence, antibiotic resistance, secretion system, and CRISPR-Cas profiles. Collectively, the findings revealed the genetic mechanisms underlying the antimicrobial activities, antibiotic resistance, and pathogenicity of *Burkholderia*.

## 2. Materials and Methods

### 2.1. Burkholderia Strains Used for Genome Comparison

For this comparative genome investigation of endophytic *Burkholderia* strains, complete genome sequences from 18 well-studied endophytic and pathogenic strains were selected (Table 1), along with 33 type strains of *Burkholderia* (Table 2). The selected species included nine endophytic strains, *Burkholderia* sp. MS389 [27], *Burkholderia* sp. MS455 [28], *Burkholderia stabilis* EB159 [29], *B. cepacia* SSG [30], *B. cenocepacia* 869T2 [31], *B. cenocepacia* YG-3 [32], *B. vietnamiensis* KJ006 [33], *B. vietnamiensis* RS1 [34], and *Burkholderia* sp. LS-044 [35]. Nine pathogenic strains were selected, including five plant pathogenic strains, *B. plantarii* ATCC 43744 [36], *B. glumae* LMG 2196 [37], *B. glumae* BGR1 [38], *B. gladioli* ATCC 10248 [37], *B. cepacia* ATCC 25416 [39], and four CF opportunistic mammalian pathogens, *B. cenocepacia* J2315 [40], *B. stabilis* ATCC BAA-67 [41], *B. multivorans* ATCC BAA-247 [37], and *B. dolosa* AU0158 [37].

### 2.2. Calculation of Whole Genome Nucleotide Identity

The genome similarity of nine endophytic and nine pathogenic *Burkholderia* genomes was determined by average nucleotide identity (ANI). ANI based on pairwise comparison was calculated using OrthoANI [42]. Strains with ANI values of more than 95%, which equates to a DNA–DNA hybridization value of more than 70%, were considered to be the same species [43,44].

### 2.3. Pan-Genome Analysis

Pan-genome analysis was performed on a dataset that included 9 endophytic strains and 33 type strains of *Burkholderia* species using the Bacterial Pan-Genome Analysis tool (BPGA) [45]. The COG (Clusters of Orthologous Groups) distribution was analyzed, and the number of core genome and pan-genome entities were plotted according to an exponential decay function or a Heaps’ power law function. The *Burkholderia* type strain database was constructed based on the pan-genome of 33 type strains. A core genome is the set of genes shared by all strains, and the accessory genome is considered the set of genes that are found in more than two (but not all) strains. Finally, genes not shared with any other strain are considered unique, and the set of nonhomologous genes within the tested genomes constitute the pan-genome.

### 2.4. Identification of the Genes Associated with the Production of Antimicrobial Compounds and Siderophores

Secondary metabolites and antibiotic-production-related genes were identified using antiSMASH [46]. Local BLAST analysis of protein sequences was employed to further confirm the related putative antibiotics-producing genes; 40% identity was used as the cutoff (BLASTP, e-value < 1 × 10^−5^) [47]. Protein-coding regions were visually compared using Easyfig, and genes were searched against the non-redundant database via BLASTx search [48]. 

### 2.5. Identification of the Genes Associated with Toxin Production, Virulence-Related Features, and Antibiotic Resistance

The protein sequences of toxin production and virulence-related genes were searched with local BLAST: e-value < 1 × 10^−5^, identity > 40% [47]. The resistance gene identifier in the comprehensive antibiotic resistance database (CARD) was used to predict genes presumed to confer antibiotic resistance [49]. Identification of integrons and their components was performed with Integron-Finder [50].

### 2.6. Identification of Secretion Systems

MacSyFinder was employed to investigate the occurrences of secretion systems in the *Burkholderia* genomes [51]. The available secretion systems were searched with a maximum e-value of 1.0, a maximum independent e-value of 0.001, and a minimum profile coverage of 0.5.

### 2.7. Identification of CRISPR-Cas System

CRISPRCasFinder was used to search for CRISPR direct repeats and spacer sequences [52]. The results, with an evidence level ≥ 1, are included in this study. The spacer sequences were subsequently processed using CRISPRTarget to predict the most likely targets of CRISPR RNA; a cut-off score of 20 was applied [53].

## 3. Results

### 3.1. Whole-Genome Nucleotide Identity

Relatedness between the endophytic and pathogenic strains of *Burkholderia* is shown in Figure 1. Strain MS455 is most closely related to endophytic strain EB159, which was isolated from mountain-cultivated ginseng [29]. Strain MS389 had the highest ANI values (95.01%) to *B. cenocepacia* YG-3, which is an endophytic strain isolated from poplars grown in a composite mine tailing [32]. However, MS389 and YG-3 shared 94.22% and 94.34% ANI values with *B. cenocepacia* type strain ATCC J2315, which suggests the taxonomic position of MS389 and YG-3 requires further investigation. In addition, the taxonomic position of the vetiver grass root isolate, *B. cenocepacia* 869T2, is questionable because of relatively low genome relatedness to any of the tested strains, including type strain *B. cenocepacia* ATCC J2315. Endophytic strain *B. cepacia* SSG and *Burkholderia* sp. LS-044 shared high ANI values (>98%) with the type strain *B. cepacia* ATCC 25416, which indicated that LS-044 is a potential member of *B. cepacia*. Plant-pathogenic strain *B. plantarii* ATCC 43744 showed limited relatedness (ANI values < 92%) to any tested strain, including *B. glumae* strains LMG 2196 and BGR1, even though both species cause similar diseases (blight) in rice. The CF isolates *B. multivorans* ATCC BAA-247 and *B. dolosa* AU0158 also shared limited relatedness (ANI values < 90%) to any tested strain. These data exemplify the arduous nature of differentiation between endophytic and pathogenic *Burkholderia* at the species level using ANI. 

### 3.2. Pan-Genomic Analysis

The pan-genome analyses of thirty-three type strains of *Burkholderia* and nine endophytic-strain genomes were conducted with comparative genomics. Overall, 34,686 putative protein-coding genes were identified, constituting the pan-genome of these 42 *Burkholderia* strains. A core–pan plot shows that the pan-genome is likely to be extended if more genomes are added to the analysis; hence, the pan-genome is considered to be “open” (Figure 2). The core genome curve levels off, which indicates the addition of more genomes to the analysis would likely not change the core genome size with any significance. The core genome followed a steep slope, reaching a minimum of 1249 gene families after the 42 genomes were added. Among the 34,686 pan-genome genes, 1249 were highly conserved across the included genomes, constituting the core genome (Figure 3). The number of unique genes ranged from 162 to 1274. A COG analysis of core, accessory, and unique genes revealed that many of the core genes are involved in amino acid transport and metabolism.

The pan-genome of the 33 type strains of *Burkholderia* was constructed as a database including their core, accessory, and unique genes in order to identify the genes related to endophytic behaviors. Overall, 31,786 putative protein-coding genes were identified as pan-genome, constituting the *Burkholderia* type strain database, which included nine well-studied endophytic strains. In total, 2506 core, 6183 accessory, and 4858 unique genes were identified in the pan-genome of the nine endophytic strains. To assess the genetic diversity of the endophytic strains, the 2506 core and 6183 accessory genes were compared with the *Burkholderia* type strain database. While there were no genes identified as unique among the 2506 core genes, there were 224 unique genes discovered within the 6183 accessory genes, indicating these genes only occur in the tested endophytic bacteria; thus, they are possibly related to the endophytic behavior of these organisms. Based on COG analysis, transcription and cell-wall/membrane/envelope biogenesis are enriched in the endophytic behavior-related genes. Among the 224 endophytic behavior-related genes, 210 are harbored in two of the nine strains. The remaining 14 genes were identified in more than two strains, which are considered to be more closely related to endophytic behavior, including genes encoding hypothetical protein, EAL domain-containing protein, H-NS histone family protein, and genes involved in fimbria biosynthesis (Figure 4).

### 3.3. Production of Antimicrobial Compounds and Siderophores

A comparison of antimicrobial-production-related gene loci is shown in Figure 5. Notably, occidiofungin—an antifungal agent produced by *Burkholderia contaminans* strain MS14, which has significant antifungal activity against several plant and human pathogens [54]—and the *ocf* gene cluster were present in the genome of strain MS455 [28]. The 58 kb *ocf* gene cluster of *B. contaminans* MS14 containing the 16 genes responsible for occidiofungin biosynthesis was also analyzed. As expected, the majority of occidiofungin biosynthesis genes, including 15 genes of regulatory gene *ambR1* and gene *ocfA* to gene *ocfN*, were present in endophytic strains EB159, KJ006, and RS1 (Figure 6). The regulatory gene *ambR2* was absent from the *ocf* gene cluster of EB159, and both MS455 and KJ006 contained an additional uncharacterized gene between the two regulatory genes *ambR1* and *ambR2*. These results suggest that strains MS455, KJ006, and EB159 may produce variants of the MS14 occidiofungin. Pyrrolnitrin is a potent antimicrobial compound produced by the *Burkholderia*, *Pseudomonas*, *Enterobacter*, and *Myxococcus* genera [55]. Several *B. cepacia* complex species have been reported to produce pyrrolnitrin [56]. The endophytic strains MS389, MS455, EB159, SSG, 869T2, YG3, and LS-044 and plant-pathogenic strain 25416 were found to harbor the biosynthesis gene of pyrrolnitrin, whereas the biosynthesis genes of antifungal compound pyoluteorin [57], an antibiotic that suppresses plant diseases caused by the plant pathogen *Pythium ultimum*, were not found in any of the genomes. The biosynthesis genes of lipopeptide AFC-BC11 [58] were found in the genomes of both endophytic and pathogenic bacteria, including MS389, EB159, YG-3, LS-044, 25416, and J2315. The plant-protective metabolite cepacin A [59] was found only in the endophyte strains KJ006 and RS1.

The biosynthesis genes of siderophore, pyochelin [60], and ornibactin [61] were extensively distributed among the studied genomes compared with the metal-chelating compound fragin [62], whose biosynthesis genes are harbored in pathogenic strains 43733 and J2315. The genetic loci responsible for the production of the antibacterial compounds sulfazecin [63] and bactobolin [64] were not found in the analyzed genomes but appeared in those of the plant-pathogenic strains 10248, 2196, and BGR1. Rhizomide [65], a lipopeptide compound, was predicted to be produced by pathogenic strain 43733, and genes for the production of the antibiotic icosalide [66] were identified in the strain 10248 genome. However, neither of these two compounds was predicted to be produced by any of the endophytic strains included in the study. 

### 3.4. Toxin Production and Virulence-Related Features

Phytotoxins and virulence-related features were compared and are shown in Figure 7. Genes involved in the biosynthesis of toxoflavin [67,68], the most important and studied phytotoxin of phytopathogenic *Burkholderia* species, were compared in the tested strains. The plant-pathogenic strains 10248, BGR1, and 2196 harbored the intact *tox* operon. Tropolone regulation genes were more commonly possessed in the studied genome. A two-component system containing three genes (*troR1*, *troK*, and *troR2*) [69] was suggested to control the production of tropolone and identified in endophytic strains MS389, 869T2, YG-3, and LS-044, as well as in pathogenic strains 10248, BGR1, 2196, 43733, J2315, 25416, and BAA-67. Rhizobitoxine produced by the pathogenic *Burkholderia andropogonis* is responsible for leaf chlorosis [70]. The genes involved in rhizobitoxine production (*rtxa*, *rtxc*, *rtxd*) were identified only in the genome of 43733. Malleilactone [71], a polyketide synthase-derived virulence factor produced by *Burkholderia pseudomallei*, was found in the genome of EB159. The biosynthesis gene of rhizoxin, a potent antimitotic macrolide that plays a key role in rice seedling blight [72], and bongkrekic acid, a highly unsaturated tricarboxylic fatty acid that causes lethal food poisoning [73], were not found in any of the genomes. The *pehA* gene, which encodes endopolygalacturonase, acts as a major virulence factor in onion pathogenicity and is involved in onion maceration in *B. cepacia* [74]. In addition to *pehA*, *pehB*, which encodes an isoform of endopolygalacturonases, was also investigated in *B. glumae*. However, the virulence functions of *pehA* and *pehB* in *B. glumae* were not confirmed because of a probable functional redundancy between the two isozymes [75]. In this study, *pehA* was identified in 10248, BGR1, 2196, 43733, 25416, and LS-044, whereas *pehB* was found only in pathogenic strains 10248, BGR1, and 2196. Lipopolysaccharide (LPS), produced by *B. cenocepacia*, has an important role in animal and human diseases [76]. The gene cluster of the O antigen, which is associated with LPS production [76], was analyzed in the studied genomes. The intact gene cluster responsible for O antigen biosynthesis was found only in the CF patient strain J2315. Overall, these results suggest that endophytic isolates harbor fewer virulence-related genes in their genomes compared with pathogenic isolates.

### 3.5. Antibiotic Resistance

The distribution of resistance genes was investigated using the CARD resistance database. Hundreds of antibiotic resistance genes (ARGs) were identified in the studied genomes. In terms of potential resistance mechanisms, resistome genes and components were categorized into six categories: antibiotic efflux, antibiotic inactivation, antibiotic target alteration, antibiotic target protection, antibiotic target replacement, and reduced permeability to antibiotics [49]. The number of ARGs ranged from 107 to 178, with the lowest value in strain BGR1 and the highest value in strain 25416 (Table 3). Non-repetitive ARGs within each of the genomes were then compared with the resistomes (Figure 8). The resistance mechanisms were similar in the analyzed genomes, excluding the antibiotic efflux systems. Efflux systems can modulate broad-spectrum antibiotic resistance, as well as resistance to specific antimicrobial compounds. Almost all species have genes that encode different types of antibiotic efflux pumps, including the MFS (major facilitator superfamily), the ABC (ATP-binding cassette) family, the RND (resistance nodulation division) family, the MATE (multidrug and toxic compound extrusion) family, and the SMR (small multidrug resistance) family, which confer resistance to macrolide antibiotics, fluoroquinolone, aminoglycoside antibiotic, cephalosporin, glycylcycline, and penam [49,77]. These results suggest endophytic and pathogenic bacteria share relatively similar profiles with antimicrobial resistance elements. However, certain genes were not evenly distributed in the analyzed genomes. The *amrA*, *amrB*, and *OprA* genes, part of the AmrAB-OprA MDR efflux pump responsible for aminoglycoside antibiotics, were not found in plant-pathogenic strains 43733, 2196, BGR1, or 10248. This suggests that resistance to aminoglycoside antibiotics in these plant-pathogenic strains may be due to the contributions of other efflux pumps. The *mdtN* and *mdtP* genes, which encode the MFS antibiotic efflux pump that provides resistance to acridine dye and nucleoside antibiotics, were found only in strains SSG and LS-044. The *abes* gene, which encodes resistance to macrolide and aminocoumarin, was demonstrated by pathogenic strains 47377, 10248, and BAA-67 but not by the endophytic strains analyzed in this study. There were also many sporadically distributed resistance (SDR) genes among the analyzed strains. Previous studies have shown that some genomes in the genus *Providencia* harbor integrons, which contribute to SDR [78]. Integrons are a major type of genetic element responsible for the spread of antibiotic resistance genes. The results revealed that the genes of integrons are not conserved in the SDR genes of the studied genomes.

### 3.6. Secretion Systems

Secretion systems are essential for bacterial interactions with surrounding environments [51]. Each secretion system has a unique function in pathogenic and beneficial plant-microbe interactions. An investigation of the occurrences of secretion systems among the studied genomes was conducted using the MacSyFinder. Genes involved in type I–VI secretion systems (T1SS-T6SS); the flagellar apparatus; and the general secretion (Sec) and twin-arginine translocation (Tat) systems were shared by some of the analyzed strains (Table 4). In general, genes involved in Sec and Tat systems and the flagellar apparatus were highly conserved among the *Burkholderia* strains. T2SS and T5SS were also highly conserved in all of the strains. T6SS, which transports effector proteins from one bacterium to another in a contact-dependent manner [79], was harbored in all analyzed strains; however, the number of T6SS gene clusters varied. Plant pathogens usually possess two to four potential functional T6SS clusters, while most endophytic bacteria contain between one to three clusters. The number of T1SSs in the analyzed strains varied, ranging from 0 to 4, with the highest values seen in strains EB159 and 869T2 and the lowest values seen in strains MS389, MS455, SSG, KJ006, RS1, 2196, 25416, and AU0158. T3SS was widely distributed in most of the endophytic and pathogenic strains but was absent from strains SSG, LS-044, and 25416. In terms of T4SS, the majority of the bacteria harbored the T4SS-excluding pathogenic strain AU0158 and endophytic strains MS389, SSG, YG-3, and RS1. These results indicate that the endophytic strains of *Burkholderia* included in this study harbored all secretion systems, including T3SS and T4SS; however, neither T3SS nor T4SS was found in previous investigations of other endophytes [79,80,81].

### 3.7. Prediction of CRISPR/Cas System

The CRISPR-Cas system is a bacterial immune system that protects bacteria from invasions from bacteriophages or foreign plasmid DNA. Recent studies have indicated that the CRISPR-Cas system is associated with virulence formation in pathogenic bacteria, acting as a barrier to horizontal gene transfer [82]. Remarkably, out of the 17 *Burkholderia* genomes analyzed, the intact CRISPR-Cas system was identified only in plant-pathogenic strain 43733 (Table 5). The CRISPR motif was found in the other 17 strains but not the Cas protein. An assumption was made that these CRISPRs could not silence foreign DNA because of a deficit in the associated Cas proteins [83]. It has been suggested that the orphan CRISPR arrays are likely remnants of previously functional CRISPR–Cas systems [83]. To further compare the CRISPR-Cas system between the endophytic and pathogenic bacteria, the CRISPR spacer sequence was analyzed using CRISPRTarget. Spacer sequences present in the endophytic *Burkholderia* genomes were identified that shared significant homology to those found in the plasmids and/or phages of the genera *Burkholderia*, *Xanthomonas*, *Sinorhizobium*, and *Mannheimia*. 

## 4. Discussion

There is a significant lack of knowledge regarding the characterization of the endophytic bacteria of *Burkholderia*. In the present comparative genome study of endophytic and pathogenic strains, we identified genetic determinants and genotypes associated with endophytic bacteria, including genes related to endophytic behavior, multiple antimicrobial products, a lack of virulence-related genes, the distribution of antibiotic resistance genes, secretion systems, and CRISPR-Cas systems.

ANI is one of the most reliable ways to identify genome-relatedness. In this study, endophytic strains *Burkholderia* sp. MS389, *Burkholderia* sp. MS455, *B. cenocepacia* YG-3, and *B. cenocepacia* 869T2 shared less than 95% ANI values with any other strains, which suggests that these endophytic strains of *Burkholderia* either remain to be further characterized, are classified incorrectly, or possibly represent a novel species. There has been a revision in the taxonomy of *Burkholderia* because of its heterogeneity, particularly in *Burkholderia cepacia* complex (Bcc). Recently, 116 Bcc strains were reclassified into 36 clusters, 22 known species, and 14 putative novel species [84]. Our genome-relatedness analysis results provide further evidence of the chaotic taxonomy situation of the genus *Burkholderia*. What is more interesting is that some pathogenic strains share high genome-relatedness with endophytic bacteria. For example, *B. epacian* 25416 shares 98.52% and 98.58% ANI values with *B. epacian* SSG and *Burkholderia* sp. LS-044, respectively, and J2315 shares 94.22% and 94.34% ANI values with MS389 and YG-3, respectively. These results suggest that a whole-genome comparison is not powerful enough evidence to distinguish the endophytic and pathogenic strains of *Burkholderia*.

The pan-genome of nine endophytic strains and thirty-three type strains was quite large and considered “open”, which indicates strains of the genus *Burkholderia* have an extensive adaptive capacity. No significant difference appeared between the accessory and unique genes of the endophytic strains, implying that the endophytes share limited characteristics at a general genetic level. Intriguingly, when 2506 core genes and 6183 accessory genes of the endophytic strains were compared with the *Burkholderia* type strain database, 0 and 224 genes were identified as endophytic behavior genes, respectively. Previous comparative studies have demonstrated that certain genes might be associated with endophytic behavior [85,86]. However, those studies were mainly focused on the comparison of endophytic bacteria between different genera rather than species. There is no doubt that genome similarity could be lower when comparing different genera. Therefore, the restricted endophytic behavior genes might be identified within a comparison scope on the species level within the *Burkholderia* genus. In this study, 14 genes were predicted as endophytic behavior genes based on their distribution among the endophytic strains, such as the biosynthesis genes of fimbriae, genes that encode EAL domain-containing protein, and genes that code for the H-NS histone family protein. It has been established that the bacterial-surface-associated structures, fimbriae, play a role in the adhesion of bacteria to host cells (including plants and animals) and the formation of biofilms [87]. The presence of fimbriae was suggested as an important component in colonizing plant roots [88]. The EAL-domain-containing protein is a ubiquitous signal transduction protein domain in bacteria and is involved in signaling mechanisms through the hydrolysis of global secondary messenger cyclic-di-GMP [89]. The presence of this EAL-domain-containing protein indicates it most likely occurs when endophytes penetrate and contact the plant host. This is consistent with previous studies showing that molecular-signal-mediated crosstalk is essential to establishing plant–microbe interactions [90]. The gene coding for histone-like nucleoid-structuring (H-NS) protein was also found in endophytic bacteria. H-NS plays a dual role in structuring DNA and in regulating transcription by acting as a pleiotropic regulator in response to environmental changes [91]. English et al. indicated that the plant-growth-promoting bacterium *Enterobacter cloacae* UW5 increases root colonization and outcompeted the wild-type strain in a direct competition assay via the overexpression of *hns* [92]. Therefore, the appearance of H-NS protein in the endophytic bacteria is not surprising. Although these genes were identified in relation to endophytic behavior, the predicted functions of the genes and their roles may only be associated with one possible dimension of plant–endophyte interactions. It is probable that each individual endophyte may utilize more unique genes to function in establishing an endophytic association with various host plants. Nevertheless, possible functions of the identified 14 genes in endophytic behavior require further characterization via mutagenesis analyses. 

Among the studied strains, endophytic strains have a wide range of common antimicrobial biosynthesis gene clusters within their genomes, including occidiofungin, pyrrolnitrin, cepacin A, and AFC-BC11, supporting the theory of the possible production of multiple antimicrobial agents that contribute to their plant-growth-promoting activity by inhibiting soil-borne plant pathogens. Furthermore, these antimicrobial biosynthesis gene clusters are often not present in the genome of pathogenic strains, especially in occidiofungin and cepacin A, which were produced by most of the endophytic strains included in the study. Pyrrolnitrin and occidiofungin, two well-studied, potent antifungal metabolites [93], were only conserved simultaneously in endophytic strains MS455 and EB159. Siderophores not only act as bacteriostatic agents to inhibit the growth of pathogenic microorganism growth by depleting iron in the soil but have also demonstrated importance in virulence to animals [94,95]. Thus, it is foreseeable that the biosynthesis genes of ornibactin and pyochelin are evenly harbored in both endophytic and pathogenic strains, which is in accordance with previous studies showing that the production of siderophores is a common trait in many *Burkholderia* species, no matter what the surviving habitats of bacteria are [96]. Intriguingly, the lipopeptide rhizomide and the antibiotics sulfazecin, bactobolin, and icosalide were absent from the endophytic strains but present in the plant-pathogenic strains. It is speculated that the existence of these antimicrobial compounds is involved in bacterial competition via antibacterial effects, which favor the predominant position of the plant-pathogenic strain in the bacterial community.

As expected, toxin production and virulence-related features were mainly identified in pathogenic *Burkholderia* species. For example, the toxoflavin biosynthesis gene cluster was identified in the plant-pathogenic strains 2196, BGR1, and 10248. However, a two-component system that contains three genes (*troR1*, *troK*, and *troR2*) and controls the production of tropolone was present in both the endophytic and pathogenic strains. *B. plantarii* was the only species known to produce the phytotoxin tropolone as a virulence factor, which causes seedling blight in the field [69,97]. There was no indication that other *Burkholderia* species could produce tropolone. Many two-component systems are conserved across the *Burkholderia* genus, and it is unclear if they all share similar functions in all species [98]. Endopolygalacturonase, encoded by the gene *pehA*, the major virulence factor in onions, was found in the plant-pathogenic strains 43733, 2196, BGR1, 10248, and 25416 but was absent from the majority of endophytic strains except for LS-044. However, it has been suggested that the plant-pathogenic strain of *B. cepacia* is less likely to become a clinical strain because the presence of *PehA* in *B. cepacia* indicates potential pathogenicity and leads to onion maceration; the clinical strains of *B. cepacia* generally do not macerate onions [74,99]. In this study, *pehA* was not found in the genome of strain SSG, an endophytic strain of *B. cepacia.* However, Kong et al. demonstrated that strain SSG was able to macerate onion-scale tissue [100]. It is reasonable to postulate that other endopolygalacturonases may be produced by strain SSG, which provides the ability to macerate onions. A comparison of toxin production and virulence-related features showed that the endophytic strains had fewer virulence-related genes in their genomes, especially in MS455, which was the only endophytic strain that lacked any virulence-related features but conserved the biosynthesis gene of two well-studied potent antifungal metabolites, pyrrolnitrin and occidiofungin.

The antibiotic resistance analysis in this study showed many of the putative antibiotic resistance elements that are shared by all *Burkholderia* genomes, with some exceptions in SDR genes. Integrons, which play an essential role in the distribution of antibiotic resistance, have also been reported. Specifically, it has been reported that an integron harbored streptothricin acetyltransferase in Bcc isolates from CF patients [101]. Tseng et al. reported that class 1 integrons contributed to antibiotic resistance in several clinical Bcc complexes [102]. In this study, the density of the integrons of all endophytic and pathogenic strains was identified with Integron-Finder. However, none of the strains contained integrons that harbored SDR genes. Thus, integrons were not the sources of multidrug resistance in this analysis.

Secretion system analyses in this study showed that all endophytic and pathogenic strains contained T2SS and T5SS. T2SS and T5SS are conserved in most Gram-negative bacteria, where they transport folded proteins from the periplasm into the extracellular environment, along with the Sec or Tat secretion pathways, which transfer protein substrates across the inner membrane [79]. Thus, it is predictable that all T2SSs and T5SSs are distributed among all studied strains. According to Seo et al., T6SS can be classified into six groups based on the distribution of T6SS components in 12 *Burkholderia* plant-pathogenic strains [103]. T6SS groups 4 and 5 secrete independent bacterial virulence factors toward host plants, while T6SS group 1 is involved in bacterial competition [103]. In this study, T6SS group 1 was conserved among all the studied *Burkholderia* strains. However, T6SS group 4 was conserved among BGR1, 2196, AU0158, KJ006, YG-3, RS1, and MS389, and group 5 was only identified among plant-pathogenic strains 2196, BGR1, and 43733. The different number of T6SSs and unique T6SS groups in the strains suggests that T6SS is not only related to plant-pathogenic bacteria colonizing host plants but also contributes to the function of endophytic bacteria host interactions. Both the endophytic and pathogenic strains harbored certain numbers of T1SSs. T1SS is present in many Gram-negative bacteria and contributes to the virulence of many bacterial pathogens, including *Vibrio cholerae* and *Serratia marcescens*, which use T1SS to secrete MARTX toxin [104] and hemophore HasA [105], respectively. Nevertheless, the virulence-related factors among *Burkholderia* species via T1SS remain unclear. In terms of T3SS, some studies have shown that, in most of the identified endophytes, T6SS appears commonly, and T3SS is relatively rare [81,106]. A further suggestion has been made stating that T3SS appears more commonly in the genomes of pathogenic bacteria rather than those of endophytic bacteria in order to modulate plant responses [80,81]. However, putative T3SS was found in many endophytic and pathogenic strains in this research. This result suggests that T3SS may be involved in promoting plant growth or interactions with other pathogenic or endophytic bacteria. T4SS and T3SS play crucial roles in pathogens in the delivery of effector proteins into the plant, which leads to effector-triggered immunity [79]. T3SS and T4SS may be either absent or present in low quantity among endophytic bacteria because of the lower possibility of inducting plant defense responses [79]. Taking T3SS and T4SS together, only strain SSG lacked both secretion systems. Piromyou et al. indicated that both T3SS and T4SS are essential components for *Bradyrhizobium* sp. SUTN9-2 to colonize the roots of rice seedlings [107]. Thus, the high abundance of T3SS and T4SS in endophytic strains may indicate their presence in the colonization of the host.

All of the analyzed *Burkholderia* genomes in this study possessed CRISPR elements, but only strain 43733 encoded Cas elements along with CRISPR regions. The results of the spacer sequence analysis indicated that the studied *Burkholderia* genomes have the potential to generate immunity against bacterial phages and plasmids. Notably, the spacer sequences of strains YG-3, LS-044, J2315, and BAA-67 were matched with the plasmid sequences of *Azoarcus* sp. KH32C. *Azoarcus* sp. KH32C was isolated from soil and identified as a denitrifying and N2O-reducing betaproteobacterium. The presence of spacer sequences matching *Azoarcus* sp. KH32C plasmid sequences in these *Burkholderia* strains indicates traces of competition interactions between *Burkholderia* and *Azoarcus*. The similar spacer sequences among these four strains suggest potential migration between endophytic and pathogenic bacteria over a long period of time. The finding of high numbers of orphan CRISPRs among the *Burkholderia* genomes is remarkable. The orphan CRISPRs may be caused by rapid genetic rearrangement in the bacterial genome, a loss of functionality in CRISPR sequences via deletion, or the lifestyle of bacteria [108]. In this study, all endophytic strains harbored CRISPR arrays without Cas proteins, while plant-pathogenic strain 43733 conserved an intact CRISPR-Cas system. In addition, intact CRISP-Cas systems have been reported in other *Burkholderia* plant-pathogenic strains, including *B. glumae* PG1, *B. glumae* 3252–8, and *B. gladioli* MSMB1756 [109]. Speculation could be made regarding how the sequences of CRISPRs in the spacers of endophytic strains may be vestiges of interactions between bacterial competitors. In addition, the CRISPR-Cas interference could limit horizontal gene transfer, and the CRISPR-Cas system may not exist in bacteria whose lifestyles are symbiotic [110,111]. Hence, it is reasonable that none of the endophytic strains have a complete and active CRISPR-Cas system.

## 5. Conclusions

Based on comparative genomic analyses of endophytic bacteria and pathogenic bacteria within the *Burkholderia* genus, distinctive features of the endophytes were established, which include the genes related to endophytic behavior, antimicrobial products, and a lack of some virulence-related genes. In total, 224 genes were absent from the *Burkholderia* type strain database but identified in the endophytic bacteria, which is suggestive of a function related to endophytic behavior. In total, 14 genes were widely conserved among the 224 endophytic behavior genes, including genes that encode hypothetical proteins, EAL-domain-containing proteins, H-NS histone family proteins, and genes involved in fimbria biosynthesis. The genomes of the endophytic bacteria of *Burkholderia* conserved more antagonistic-activity-related genes than the pathogenic bacteria, such as biosynthesis genes of occidiofungin, pyrrolnitrin, and cepacin A. The genes related to virulence features, the biosynthesis gene of toxoflavin, genes that encode endopolygalacturonase, and O-antigen were rarely found among the endophytes. Despite fewer antibiotic resistance genes occurring only in the endophytic and pathogenic bacteria, the majority of resistome genes were relatively evenly distributed among the studied strains. The wide identification of genes for protein secretion systems was well characterized. Notably, T3SS and T4SS, which are not believed to be ubiquitously distributed among endophytes, were conserved in the genomes of the studied *Burkholderia* endophytes. Unlike pathogenic bacteria that harbored the intact CRISPR-Cas system, endophytic bacteria prevalently had questionable CRISPR-Cas systems along with an orphan CRISPR array. In conclusion, this study analyzed the genetic diversity and distinct features of endophytic bacteria within the *Burkholderia* genus and highlighted the potential of endophytes as biocontrol agents.

## Figures and Tables

**Figure 1 microorganisms-12-00100-f001:**
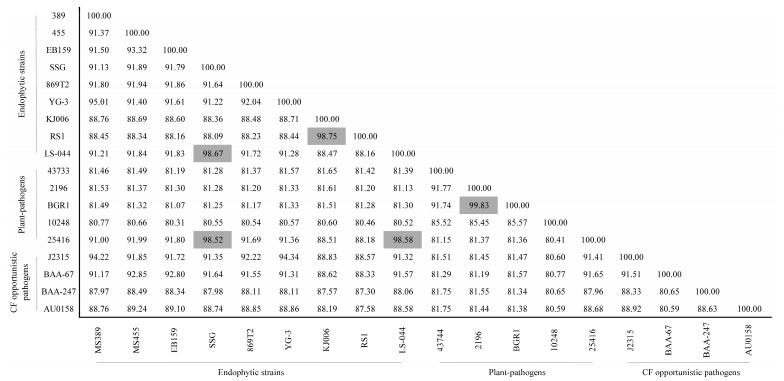
Average nucleotide identity (ANI) pairwise comparisons of endophytic and pathogenic strains of *Burkholderia*. Gray area (ANI > 96%) indicates the analyzed strains could be considered to be the same species.

**Figure 2 microorganisms-12-00100-f002:**
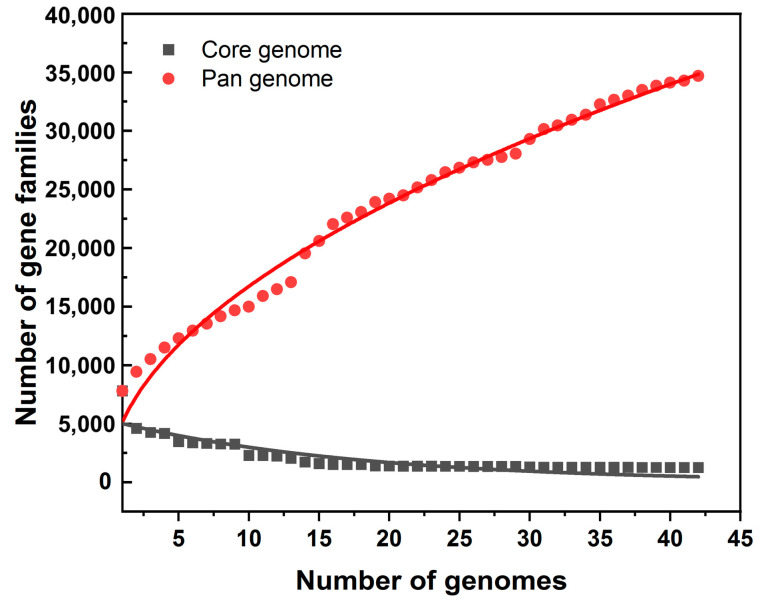
Core–pan-genome plot of 9 endophytic strains and 33 type strains of *Burkholderia*. The total number of core genomes is shown in black, and the total number of pan-genomes is shown in red.

**Figure 3 microorganisms-12-00100-f003:**
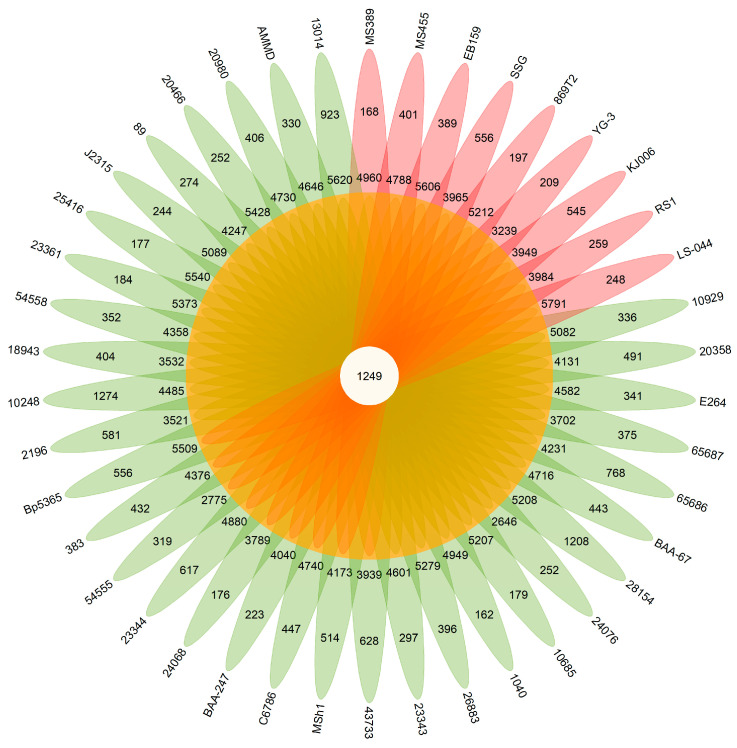
Pan-genome analysis of 9 endophytic strains and 33 type strains of *Burkholderia*. Flower diagram of pan-genome representing the core, accessory, and unique genes. Red ovals represent endophytic stains; green ovals represent type strains of *Burkholderia*. The center portion is the number of core genes. Numbers in the middle portions of each oval indicate the accessory genes. Numbers in distal portions of each oval indicate the unique genes.

**Figure 4 microorganisms-12-00100-f004:**
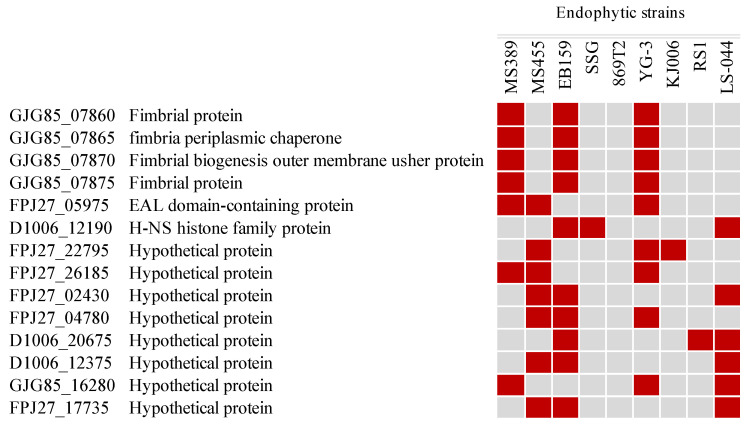
Putative endophytic behavior genes distributed in endophytic strains of *Burkholderia*. Red boxes represent presence; gray boxes represent absence.

**Figure 5 microorganisms-12-00100-f005:**
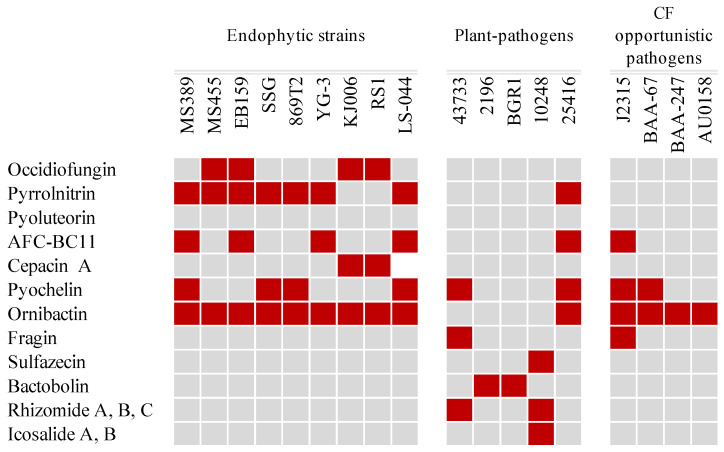
Antibiotic and siderophore genes distributed in endophytic and pathogenic strains of *Burkholderia*. Red boxes represent presence; gray boxes represent absence.

**Figure 6 microorganisms-12-00100-f006:**
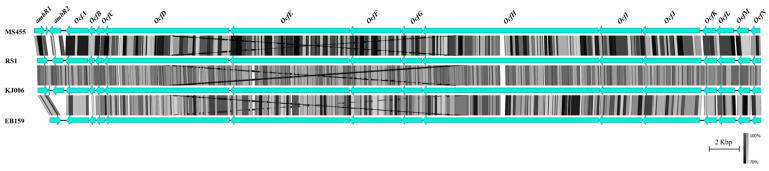
Occidiofungin biosynthesis locus genetics. The 58 kb occidiofungin biosynthesis locus identified in 4 endophytic strains of *Burkholderia*.

**Figure 7 microorganisms-12-00100-f007:**
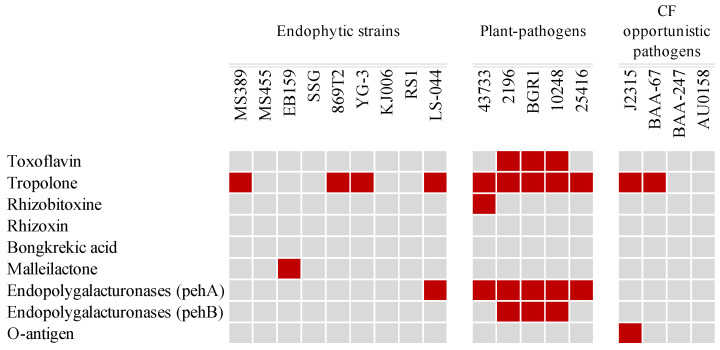
Toxin production and virulence-related features of genes distributed in endophytic and pathogenic strains of *Burkholderia*. Red boxes represent presence; gray boxes represent absence.

**Figure 8 microorganisms-12-00100-f008:**
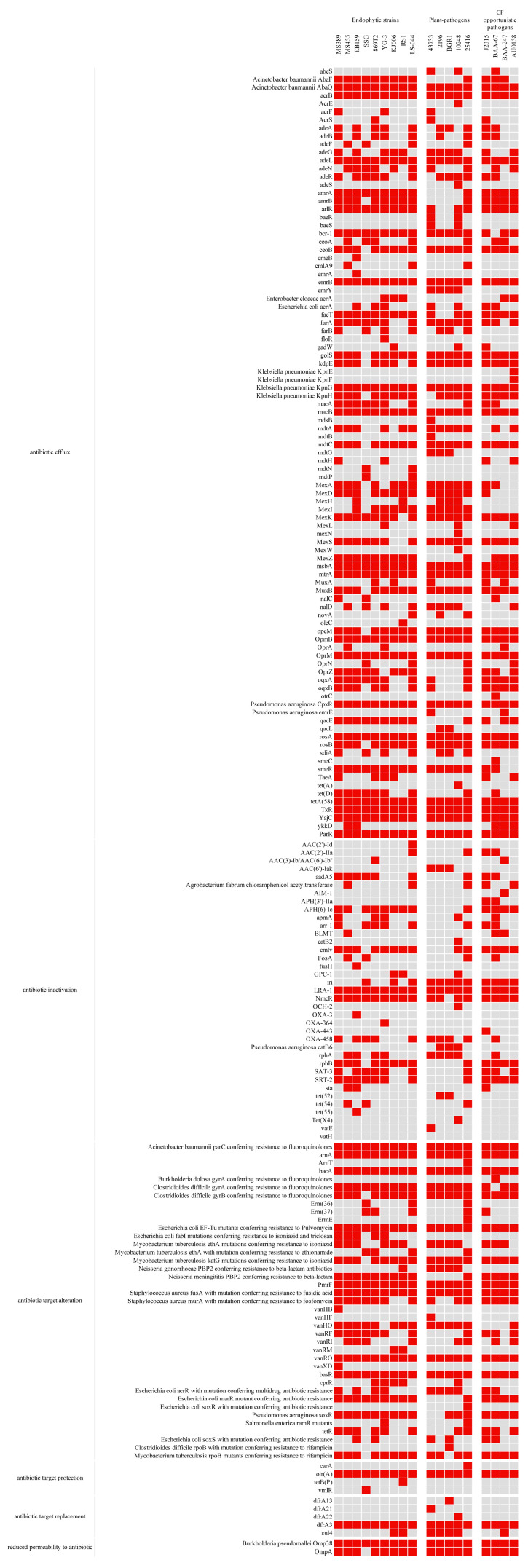
Resistome genes distributed in endophytic and pathogenic strains of *Burkholderia*. Red boxes represent presence; gray boxes represent absence.

**Table 1 microorganisms-12-00100-t001:** List of endophytic and pathogenic strains of *Burkholderia* used in comparative genomic analysis.

	Strain	# Chr	# Plasmid	Size (bp)	GC%	Isolation Source	Accession Number
Endophytic strains	*Burkholderia* sp. MS389	3	-	7,746,250	66.73	Soybean plant	GCA_016899425
	*Burkholderia* sp. MS455	3	1	8,055,789	66.06	Soybean plant	GCA_016899445
	*B. stabilis* EB159	-	-	9,093,480	66.00	Mountain cultivated ginseng	GCA_004125865
	*B. cepacia* SSG	-	-	8,571,737	66.94	Boxwood leaves	GCA_012510315
	*B. cenocepacia* 869T2	-	-	7,979,184	67.07	Vetiver grass roots	GCA_000705535
	*B. cenocepacia* YG-3	3	-	8,036,463	66.81	Poplar tree	GCA_003966315
	*B. vietnamiensis* KJ006	3	1	6,629,912	67.18	Rice root	GCA_000262695
	*B. vietnamiensis* RS1	-	-	6,542,727	67.61	Tuberous roots of sweet potato	GCA_003203575
	*Burkholderia* sp. LS-044	-	-	8,782,274	66.52	Rice root	GCA_004803625
Plant pathogens	*B. plantarii* ATCC 43733	2	1	8,081,051	68.55	Diseased rice sheath	GCA_001411805
	*B. glumae* LMG 2196	2	2	6,820,727	68.18	Diseased rice grain	GCA_000960995
	*B. glumae* BGR1	2	4	7,284,636	67.93	Diseased rice grain	GCA_000022645
	*B. gladioli* ATCC 10248	2	3	8,899,459	67.63	Diseased gladiolus corms	GCA_000959725
	*B. cepacia* ATCC 25416	3	1	8,605,945	66.61	Rotten onion	GCA_001411495
CF opportunistic pathogens	*B. cenocepacia* J2315	3	1	8,055,782	66.90	Sputum of cystic fibrosis patients	GCA_000009485
	*B. stabilis* ATCC BAA-67	3	-	8,527,947	66.42	Sputum of cystic fibrosis patients	GCA_001742165
	*B. multivorans* ATCC BAA-247	3	-	6,322,746	67.24	Sputum of cystic fibrosis patients	GCA_000959525
	*B. dolosa* AU0158	3	-	6,409,095	67.01	Sputum of cystic fibrosis patients	GCA_000959505

**Table 2 microorganisms-12-00100-t002:** List of type strains of *Burkholderia* used in pan-genome analysis.

Strain	# Chr	# Plasmid	Size (bp)	GC%	Isolation Source	Accession Number
*B. aenigmatica* LMG 13014	-	-	8,897,342	65.95	Contaminant of hand cream	GCA_902499175
*B. ambifaria* AMMD	3	-	7,528,567	66.77	Rhizosphere of pea	GCA_000203915
*B. anthina* LMG 20980	-	-	7,608,177	66.74	Rhizosphere of a house-plant	GCA_902498995
*B. arboris* LMG 24066	-	-	8,271,305	66.84	Morris Arboretum soil	GCA_902499125
*B. catarinensis* 89	-	-	8,137,374	66.46	Native grassland soil	GCA_001883705
*B. cenocepacia* J2315	3	1	8,055,782	66.9	Sputum of cystic fibrosis patients	GCA_000009485
*B. cepacia* ATCC 25416	3	1	8,605,945	66.61	Rotten onion	GCA_001411495
*B. contaminans* LMG 23361	4	-	10,352,616	65.68	Milk of dairy sheep with mastitis	GCA_001758385
*B. diffusa* CCUG 54558	-	-	7,111,827	66.25	Sputum of cystic fibrosis patients	GCA_008802145
*B. dolosa* LMG 18943	-	-	6,079,078	66.97	Sputum of cystic fibrosis patients	GCA_902499135
*B. gladioli* ATCC 10248	2	3	8,899,459	67.63	Diseased gladiolus corms	GCA_000959725
*B. glumae* LMG 2196	2	2	6,820,727	68.18	Diseased rice grain	GCA_000960995
*B. humptydooensis* MSMB43	2	1	7,287,809	67.14	Bore water sample	GCA_001513745
*B. lata* 383	3	-	8,676,277	66.27	Forest soil	GCA_000012945
*B. latens* CCUG 54555	-	-	7,103,502	66.82	Sputum of cystic fibrosis patients	GCA_008802135
*B. mallei* ATCC 23344	-	-	5,835,527	68.49	Etiological agent of glanders	GCA_000011705
*B. metallica* LMG 24068	-	-	7,532,498	67.04	Sputum of cystic fibrosis patients	GCA_902499065
*B. multivorans* ATCC BAA-247	3	-	6,322,746	67.24	Sputum of cystic fibrosis patients	GCA_000959525
*B. oklahomensis* C6786	2	-	7,135,022	67.07	Deep leg wound heavily contaminated with soil	GCA_000959365
*B. paludis* MSh1	-	-	8,633,651	67.08	Southeast Pahang tropical peat swamp forest soil	GCA_000732615
*B. plantarii* ATCC 43733	2	1	8,081,051	68.55	Diseased rice sheath	GCA_001411805
*B. pseudomallei* ATCC 23343	-	-	7,037,422	68.27	Etiological agent of melioidosis	GCA_001182285
*B. pseudomultivorans* LMG 26883	-	-	7,396,511	66.99	Human respiratory specimens	GCA_902499075
*B. puraquae* CAMPA 1040	-	-	8,098,134	66.59	Hemodialysis water	GCA_002099195
*B. pyrrocinia* DSM 10685	3	1	7,961,346	66.46	Environmental soil	GCA_001028665
*B. seminalis* LMG 24067	-	-	7,928,753	67.07	Sputum of cystic fibrosis patients	GCA_902499165
*B. singularis* LMG 28154	-	-	5,538,112	64.34	Sputum of cystic fibrosis patients	GCA_900176645
*B. stabilis* ATCC BAA-67	3	-	8,527,947	66.42	Sputum of cystic fibrosis patients	GCA_001742165
*B. stagnalis* CCUG 65686	-	-	8,125,942	66.94	Environmental soil	GCA_008802125
*B. territorii* CCUG 65687	-	-	6,834,211	66.55	Environmental groundwater	GCA_008802115
*B. thailandensis* E264	2	-	6,723,972	67.63	Rice field soil	GCA_000012365
*B. ubonensis* LMG 20358	-	-	7,692,957	67.24	Surface soil of roadside	GCA_902499185
*B. vietnamiensis* LMG 10929	3	1	6,930,496	66.83	Rice rhizosphere soil	GCA_000959445

**Table 3 microorganisms-12-00100-t003:** Summary of resistance mechanisms distributed in endophytic and pathogenic strains of *Burkholderia*.

	Name	Antibiotic Efflux	Antibiotic Inactivation	Antibiotic Target Alteration	Antibiotic Target Protection	Antibiotic Target Replacement	Reduced Permeability to Antibiotic	Total
Endophytic strains	MS389	101	13	29	1	1	13	158
	MS455	88	14	25	1	1	14	143
	EB159	98	15	28	1	1	19	162
	SSG	63	12	21	2	1	11	110
	869T2	109	14	30	1	1	13	168
	YG-3	105	12	31	1	1	12	162
	KJ006	67	10	23	1	2	10	113
	RS1	62	10	25	2	2	9	110
	LS-044	113	17	27	1	1	17	176
Plant pathogens	43733	90	10	23	2	3	14	142
	2196	68	8	19	2	2	9	108
	BGR1	65	7	21	2	3	9	107
	10248	105	13	28	1	3	20	170
	25416	108	18	32	2	1	17	178
CF opportunistic pathogens	J2315	92	18	27	1	1	10	149
	BAA-67	96	20	27	1	1	18	163
	BAA-247	68	13	20	1	2	12	116
	AU0158	74	10	25	1	1	10	121

**Table 4 microorganisms-12-00100-t004:** Summary of secretion systems distributed in endophytic and pathogenic strains of *Burkholderia*.

	Name	T1SS	T2SS	T3SS	T4SS	T5SS	T6SS	Sec	Tat	Flagellar Apparatus
Endophytic strains	MS389	-	1	1	-	15	3	1	1	1
	MS455	-	1	1	1	9	3	1	1	1
	EB159	4	1	1	3	19	2	1	1	1
	SSG	-	1	-	-	3	2	1	1	1
	869T2	4	1	1	1	10	1	1	1	1
	YG-3	2	1	1	-	13	3	1	1	1
	KJ006	-	1	1	1	12	2	1	1	1
	RS1	-	1	1	-	12	2	1	1	1
	LS-044	1	1	-	1	9	3	1	1	1
Plant pathogens	43733	2	1	1	2	6	3	1	1	1
	2196	-	1	1	1	7	3	1	1	1
	BGR1	1	1	1	2	9	4	1	1	1
	10248	1	1	1	1	10	2	1	1	1
	25416	-	1	-	1	9	2	1	1	1
CF opportunistic pathogens	J2315	2	1	1	2	8	1	1	1	1
	BAA-67	1	1	1	1	9	2	1	1	1
	BAA-247	1	1	1	2	7	2	1	1	1
	AU0158	-	1	1	-	14	2	1	1	1

**Table 5 microorganisms-12-00100-t005:** Summary of predicted CRISPR-Cas systems distributed in endophytic and pathogenic strains of *Burkholderia*.

	Name	CRISPR	Cas	Spacer
Endophytic strains	MS389	-	-	-
	MS455	3	-	2
	EB159	1	-	1
	SSG	2	-	4
	869T2	3	-	3
	YG-3	2	-	2
	KJ006	3	-	2
	RS1	2	-	1
	LS-044	6	-	8
Plant pathogens	43733	5	1	22
	2196	2	-	3
	BGR1	1	-	2
	10248	4	-	1
	25416	2	-	2
CF opportunistic pathogens	J2315	1	-	1
	BAA-67	2	-	1
	BAA-247	2	-	3
	AU0158	4	-	2

## Data Availability

Data are contained within the article.

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
