# Peer review of "Comparative Genome Analyses Provide Insight into the Antimicrobial Activity of Endophytic Burkholderia"

_microorganisms, 2024, doi:10.3390/microorganisms12010100_

Round 1

Reviewer 1 Report

Comments and Suggestions for Authors

General information:

This is an interesting manuscript addressing an important scientific issue. It is beginning to be more and more evident that the identification of microorganisms to genus level is not enough to assess if these microorganisms are pathogenic or even beneficial. This can pose a huge problem for the identification of etiological factors of bacterial diseases. Hopefully, better access to full genomic data and tools for genomic data analysis will bring us closer to finding the means for quick differentiation. The authors present a much-needed analysis of the genus Burkholderia and its genes associated with its ecological functions. The manuscript contains an insightful introduction to the presented problem followed by a clear presentation of properly selected methods. The results are easy to follow with the numerous esthetic graphics and tables. The discussion summarises these interesting results with a reference to recent literature. There are only a few minor issues that should addressed before publication. The authors should carefully check if all scientific names are written in italics. The list of references should be prepared by the journal guidelines. Finally, I would appreciate it if the authors would add the tables presenting row results to the supplementary materials. Summarizing I consider this manuscript acceptable for publication in its current form after small editorial changes.

In-text comments:

Line 2: Please do not divide words in titles.

Line 8: Add an asterisk after the corresponding author.

Line 14: Please keep italics for scientific names

Line 164: Please exchange this figure for a higher-quality version

Line 182: Please add a thousand separators to this figure

Line 570: Please move the Acknowledgement to the acknowledgment section and confirm that acknowledged people consent to be acknowledged

Line 577: I do not see how this is a conflict of interest. I understand the company did not participate in the research or its design. The presented results should not directly harm or benefit the company. Please explain that this does not influence the presented analysis's quality.

Author Response

Reviewer 1:

Line 2: Please do not divide words in titles.

Response: We apologize for the error, and it has been corrected.

Line 8: Add an asterisk after the corresponding author.

Response: We apologize for the error, and it has been corrected.

Line 14: Please keep italics for scientific names

Response: We apologize for the error, and it has been corrected.

Line 164: Please exchange this figure for a higher-quality version

Response: We appreciate the suggestion, a higher quality figure was placed.

Line 182: Please add a thousand separators to this figure

Response: We appreciate the suggestion, and thousand separators were added.

Line 570: Please move the Acknowledgement to the acknowledgment section and confirm that acknowledged people consent to be acknowledged

Response: We apologize for the error, and it has been corrected. The acknowledged people have been confirmed.

Line 577: I do not see how this is a conflict of interest. I understand the company did not participate in the research or its design. The presented results should not directly harm or benefit the company. Please explain that this does not influence the presented analysis's quality.

Response: We appreciate the comment made by the expert. We agree with the reviewer that there is no conflict of interest associated with the publication. We have changed it to “Authors have no conflict of interest to declare”.

Reviewer 2 Report

Comments and Suggestions for Authors

The manuscript presents a thorough comparative genomic analysis of endophytic and pathogenic Burkholderia strains, focusing on various aspects including average nucleotide identity (ANI), pan-genome analysis, antimicrobial compound production, toxin and virulence factor genes, antibiotic resistance, secretion systems, and CRISPR-Cas elements. The authors effectively establish distinct genetic features of endophytic Burkholderia, highlighting their potential as biocontrol agents due to their antimicrobial properties and less frequent occurrence of virulence-related genes compared to pathogenic strains. The study provides valuable insights into the genetic complexity and diversity within the Burkholderia genus, contributing to our understanding of the distinct roles and behaviors of endophytic versus pathogenic bacteria.

Please read comments below:

Check again the whole manuscript for the style of the bacterial genus and species name: Burkholderia -> Burkholderia (italic)

Line 19: "All types of secretion systems were found in the studied bacteria" is a broad statement. Consider specifying if this refers to all ‘known’ types or all types relevant to the study.

Line 19-20: Suggest rephrasing for clarity: "This includes T3SS and T4SS systems, which were previously thought to be disproportionately represented in endophytes."

Line 20-21: Clarify "questionable CRISPR-Cas systems." Do you mean systems of uncertain function or incomplete systems?

Line 22-23: The conclusion can be strengthened. Suggest: "This research not only sheds light on the antimicrobial activities that contribute to biocontrol but also expands our understanding of genomic variations in Burkholderia's endophytic and pathogenic bacteria."

Line 59-60: The list of genera could be presented more clearly. Suggestion: "The Burkholderia sensu lato group has been divided into seven genera: Burkholderia sensu stricto, Paraburkholderia, Caballeronia, Mycetohabitans, Trinickia, Robbsia, and Pararobbsia [17-20]."

Line 63, 69: remove the unnecessary period. ‘. [22].’, ‘activities. [24].’

Figure 6 and 8 are impossible to read. Please provide pictures with higher resolution and consider separate them in a horizontal view and a separate page.

Line 574-576: Please delete the part or add relevant information.

Author Response

Reviewer 2:

Check again the whole manuscript for the style of the bacterial genus and species name: Burkholderia -> Burkholderia (italic)

Response: We apologize for the error, and it has been corrected.

Line 19: "All types of secretion systems were found in the studied bacteria" is a broad statement. Consider specifying if this refers to all ‘known’ types or all types relevant to the study.

Response: We appreciate the suggestion. The sentence was revised as suggested.

Line 19-20: Suggest rephrasing for clarity: "This includes T3SS and T4SS systems, which were previously thought to be disproportionately represented in endophytes."

Response: We appreciate the suggestion. The sentence was revised as suggested.

Line 20-21: Clarify "questionable CRISPR-Cas systems." Do you mean systems of uncertain function or incomplete systems?

Response: We apologized for ambiguous description. the CRISPR-Cas system is not complete and doubtable based on our analysis. Thus, we used “questionable” and numerous published papers use “questionable” in this situation.

Line 22-23: The conclusion can be strengthened. Suggest: "This research not only sheds light on the antimicrobial activities that contribute to biocontrol but also expands our understanding of genomic variations in Burkholderia's endophytic and pathogenic bacteria."

Response: We appreciate the suggestion. The sentence was revised as suggested.

Line 59-60: The list of genera could be presented more clearly. Suggestion: "The Burkholderia sensu lato group has been divided into seven genera: Burkholderia sensu stricto, Paraburkholderia, Caballeronia, Mycetohabitans, Trinickia, Robbsia, and Pararobbsia [17-20]."

Response: We appreciate the suggestion. The sentence was rephrased.

Line 63, 69: remove the unnecessary period. ‘. [22].’, ‘activities. [24].’

Response: We apologize for the error, and it has been corrected.

Figure 6 and 8 are impossible to read. Please provide pictures with higher resolution and consider separate them in a horizontal view and a separate page.

Response: We appreciate the suggestion, a higher quality figure was placed with horizontal view.

Line 574-576: Please delete the part or add relevant information.

Response: We apologize for the error, and it has been corrected.